# Prediction of Surface Roughness of Abrasive Belt Grinding of Superalloy Material Based on RLSOM-RBF

**DOI:** 10.3390/ma14195701

**Published:** 2021-09-30

**Authors:** Ying Liu, Shayu Song, Youdong Zhang, Wei Li, Guijian Xiao

**Affiliations:** College of Mechanical and Vehicle Engineering, Chongqing University, Chongqing 400044, China; lyjfb@163.com (Y.L.); songshayu@cqu.edu.cn (S.S.); zhangyoudong@cqu.edu.cn (Y.Z.); 13452455686@163.com (W.L.)

**Keywords:** abrasive belt grinding, surface roughness prediction, radial basis function neural network, reinforcement, nickel-based superalloy

## Abstract

It is difficult to accurately predict the surface roughness of belt grinding with superalloy materials due to the uneven material distribution and complex material processing. In this paper, a radial basis neural network is proposed to predict surface roughness. Firstly, the grinding system of the superalloy belt is introduced. The effects of the material removal process and grinding parameters on the surface roughness in belt grinding were analyzed. Secondly, an RBF neural network is trained by reinforcement learning of a self-organizing mapping method. Finally, the prediction accuracy and simulation results of the proposed method and the traditional prediction method are analyzed using the ten-fold cross method. The results show that the relative error of the improved RLSOM-RBF neural network prediction model is 1.72%, and the R-value of the RLSOM-RBF fitting result is 0.996.

## 1. Introduction

Superalloy at high temperatures has excellent strength, good oxidation resistance, and thermal corrosion resistance properties. GH4169 superalloy, a specific type of metal material, is composed of iron, nickel, and cobalt, and can serve for a long time under particular stress and high temperatures of more than 600 °C [1]. However, due to the high plasticity and low thermal conductivity of superalloy, many problems have been encountered in the traditional cutting process, for example, the cutting force is too large, the work hardening phenomenon is severe, the grinding heat is high, the cutting deformation is large, and it becomes worn easily [2].

Because belt grinding is a “cold state” processing method, it has high economic efficiency and fast processing efficiency and is widely used in various manufacturing fields. Since the grinding process involves multi-edge cutting, the material removal rate is low, and the cutting head is small, so the roughness after grinding is small [3,4,5,6,7,8]. Huang et al. [9] analyzed the vibration mechanism from the perspective of dynamics and established a vibration model of impeller belt grinding, which was verified experimentally with different parameters and was found to be effective in reducing the vibration phenomenon during the grinding process with better parameters achieved in the experiments. The surface roughness after grinding with this method was 0.291~0.368 μm. Fan et al. [10] established the contact relationship between the abrasive grains of the abrasive belt and the material’s surface morphology by analyzing the abrasive belt’s surface morphology characteristics. Xiao et al. [11] used superalloy materials to carry out abrasive belt grinding experiments, performed an experimental comparison and analysis on the upper and lower end grinding faces, and studied the workpiece’s surface integrity and residual stress after grinding. A multi-abrasive particle model for abrasive belt grinding was established, and finite element software was used for simulation analysis. The influence of plastic flow on the residual stress distribution on the grinding surface was obtained. Xiao et al. [12] processed bionic drag reduction microstructures on the workpiece surface by abrasive belt grinding and carried out a numerical simulation to obtain the influence of bionic drag reduction microstructures in abrasive belt grinding on the dynamic performance of blades. Belt grinding has become a new machining method for typical complex materials such as superalloy. Due to the uneven abrasive distribution in belt grinding, the material is challenging to process. The uncertainty of abrasive particle distribution and machining errors have brought significant challenges to constructing the mathematical relationship between machining parameters and surface integrity. In addition, due to wear, machining error, and chatter in the process, it is difficult to predict the roughness of the workpiece after machining.

At present, the prediction of workpiece surface quality is a relatively hot topic. Roughness is an essential feature of surface quality, and the research on it is relatively mature. Hu et al. [13] made predictions of the roughness of the workpiece formed by fusion deposition technology and proposed a theoretical model to express the roughness distribution according to the change of the surface angle. Furthermore, in order to verify the correctness of the model, relevant experimental studies have been carried out. Surface roughness obtained the average relative error of predicted roughness, and the measured roughness was 6.25% and 5.04%, respectively. Klocke et al. [14] first conducted grinding experiments using nickel-based alloys. Secondly, they constructed a prediction model of processing parameters (grinding temperature, grinding pressure) and surface integrity based on experiments. Lu et al. [15] used the support vector machine (SVM) model optimized by the differential evolution algorithm to build a vermicular graphite cast iron roughness (*Ra*) prediction model. They accurately predicted vermicular graphite cast iron’s roughness in the machining process. The differential evolution theory is used to construct a prediction model for workpiece roughness based on the support vector machine algorithm. Through experiments, the prediction accuracy of the model was verified. The established prediction model excavated the relationship between surface roughness and machining parameters to obtain better machining parameters. Wu et al. [16] introduced the roughness correction coefficients φ1 and τ1 for plastic removal and the roughness correction coefficients φ2 and τ2 for plastic-brittle removal to establish a prediction model for roughness based on different removal methods. They solved the coefficients through grinding experiments to obtain the relationship between processing parameters on roughness and topography. Yi et al. [17] studied the surface quality of the new grinding wheel after grinding, analyzed the angle of the grinding zone and grinding parameters, and analyzed the surface quality after grinding. The optimal process parameters of surface quality were obtained. Ma et al. [18] carried out roughness experiments on mica in fast point grinding. Combining Malkin’s kinematics model and Snoeys empirical model, a modified point grinding roughness containing five grinding factors was established, which opposed some scholars’ view that the deflection angle was not related to roughness. Liu et al. [19] explored the influence of polishing pressure, abrasive concentration, abrasive particle size, and processing time on the polished microholes’ surface roughness through orthogonal tests. The results showed that the abrasive flow one-way cycle polishing is beneficial in improving the nozzle structure. The machining parameters, polishing pressure, and abrasive concentration significantly affect the polishing runner’s surface roughness. Lin and Li et al. [20] analyzed the influence of the grinding wheel’s abrasive grain characteristics and process parameters on the surface quality. They used the improved Pareto particle swarm algorithm to optimize the two parameters of production efficiency and roughness. In the grinding experiments of C-250 maraging steel, Guo et al. [21] collected force signal and acoustic signal characteristics. A long short-term memory (LSTM) network algorithm is proposed based on a time series to predict workpiece roughness. Gu et al. [22] carried out grinding experiments of bearings and established the relationship between process parameters and surface topography. The gray wolf algorithm is used to optimize the support vector machine algorithm, and the error under the conditions of the algorithm can be controlled within 10%.

With modern related intelligent algorithms, artificial intelligence and its optimization algorithms have been further applied in grinding processing. Liu and Yu et al. [23] proposed a model based on a Knowledge Deep Belief Network (KBDBN) to predict the processed surface quality, which realized accurate predictions and effectively extracted essential knowledge about the manufacturing process. By comparing the experimental results, KBDBN innovatively combines symbolic rules with the deep learning theory. Through simulation analysis, it can be determined that the network performance is better, and it has good interpretability for experimental data and strong generalization ability. The prediction model of machining roughness was established, extracting process knowledge based on accurate prediction and guiding the machining process optimization. Amamou et al. [24] used the improved neural network algorithm to predict the grinding force ratio component’s amplitude. This enhanced neural network could select the optimal input training set for training, including the critical factors and the interaction between the elements, and this could be learned and generalized. The results showed that this method performs better than the regression model, the genetic algorithm (20.32%) model, and the traditional neural network (7.84%) model. Simultaneously, the optimal training set selection improved the generalization ability of the algorithm. Prabhu [25] combined the neural network algorithm (the Levenberg–Marquardt conjugate gradient method) with the Taguchi method and the fuzzy logic method to predict the surface roughness of Carbon Nanotube (CNT) hybrid nanofluid technology. Sedighi et al. [26] used the GA theory to improve the Back Propagation (BP) algorithm and optimized the creep feed grinding (CFN) process. Pandiyan et al. [27] used the genetic algorithm based on K-Nearest Neighbor (KNN) combined with an SVM to carry out the tool wear process of inflexible tool cutting (essentially, belt grinding) and analyzed the numerical characteristics.

It can be seen from the above-mentioned literature that the uneven distribution of abrasive belt grains introduces difficulties in the prediction of surface roughness, and the intelligent algorithm with surface roughness as the main prediction object has been applied in various processing methods, and is used in grinding processing. There have been more mature applications, but there is still a lack of predictive models and error optimization analysis for grinding process parameters.

In this paper, the RLSOM-RBF (radial basis function) method is proposed for the uneven distribution of abrasive particles in belt grinding to solve the problem whereby the nonlinear relationship between process parameters and surface roughness is not easy to predict. This paper improves the self-organization map (SOM) method through reinforcement learning (RL), designs belt-grinding experiments, analyzes the effectiveness of the method, and finally, uses experimental data as training samples and the error curved surface model, the model error, training results, and changes of the parameters as standards to further verify that the proposed method can be used for surface roughness prediction of abrasive belt grinding with superalloy materials.

## 2. Prediction Model of Surface Roughness of Abrasive Belt Grinding of Superalloy Material

### 2.1. Abrasive Belt Grinding System

Abrasive belt grinding is a kind of elastic grinding. Since the carrier of the abrasive belt is composed of materials with certain elasticity, such as a cloth base, and the flexible characteristics of the adhesive and rubber contact wheel, abrasive belt grinding is a grinding and polishing compound processing technology. In addition, abrasive belt grinding has already entered the precision and ultra-precision machining ranks, and the highest precision reached is 0.1 μm.

Abrasive belt grinding is a particular form of coated abrasive with an abrasive belt. It is tensioned utilizing a tensioning mechanism and driven by a drive wheel to move at high speed. Then, according to the shape and processing requirements of the workpiece, a particular amount of pressure is applied. The belt is in contact with the surface of the workpiece through the entire process of grinding. By setting the grinding force, the linear speed of the abrasive belt, the feed speed, and other related parameters, the workpiece is ground and polished to obtain a good surface roughness.

Many factors affect the surface roughness of abrasive belt grinding. According to relevant theoretical formulas, the relationship between surface roughness (*Ra*) and each grinding parameter can be expressed as:(1)$$Ra=KF^{\beta }v_{f}^{\gamma }v_{s}^{\delta },$$

Among them, *K* is the scale factor, *F* is the grinding pressure, *v_f_* is the feed rate, and *v_s_* is the linear velocity of the abrasive belt, while *β*, *γ*, and *δ* are coefficients related to grinding parameters.

It can be seen from the appealing formula that the roughness value is related to the grinding pressure, the belt speed, and the feed speed, and these three parameters are related to the problem of being highly nonlinear and highly coupled.

### 2.2. Radial Basis Function Center Training Method for Strengthening Self-Organizing Mapping

The radial basis function (RBF) neural network is a neural network technology that interpolates in a high-latitude space and usually consists of an input layer, a hidden layer, and an output layer. There are three parameters that RBF needs to learn to determine, which are the center of the radial basis function, the variance, and the weights between the output layers of the hidden layer domain. According to the different radial basis function center methods, the radial basis function neural network has a variety of learning methods, such as the random selection center method, the self-organizing map selection center method, the supervised selection center method, and the orthogonal least-squares method. Traditional learning methods will cause RBF neural networks to fall into local optimal solutions and other related problems, so this paper combines reinforcement learning (RL) and a self-organizing map (SOM).

RL-SOM (Reinforcement Learning with Self-Organization Map) uses the correction amount of the clustering center of neurons as a reward and judges the size of the reward according to the quality of the superalloy abrasive belt after grinding. When training with random samples, a greater amount of correction for neurons with greater competitiveness enhances the competitiveness of neurons and stimulates competition among neurons based on environmental feedback. At the same time, it is necessary to avoid the tendency of neurons to be consistent during the match and fall into a locally optimal solution. RL-SOM introduces randomness in a probabilistic way in the process of assigning rewards. Neurons with greater competitiveness receive positive tips (a greater probability of a higher amount of correction in the sample) but do not receive positive rewards. The introduction of randomness enhances the algorithm’s global search ability. Finally, neurons continue to be trained. They learn and explore based on environmental feedback to seek to win the competition. Overall, the RL-SOM method generates the RBF surface roughness prediction model. The realization of the radial base center has two main steps: First, the RL-improved SOM method is used for competition. The reward mechanism constructs a three-layer competitive neural network predictive sample clustering model. It then uses the improved stochastic gradient enhancement method [17] to train model parameters, update the clustering center, and finally, use the clustering center of the most competitive neuron as the base center.

The parameter training of the RL-SOM clustering model is an essential link. RL-SOM mainly trains the parameter model through the improved stochastic gradient reinforcement method. The primary training process is shown in Figure 1.

In order to enhance the ability to search for the optimal global solution, RL-SOM has made certain modifications *r*. The improved reward formula is as follows:(2)$$r_{j}=\{\begin{array}{c} h_{j};q_{j}=1\\ -h_{j};q_{j}=0 \end{array},\;h_{j}=\exp\left(\frac{\left\|c_{j}-c^{*}\right\|}{2\sigma^{2}}\right)$$

The *h_j_* formula is the update area; *σ* is the effective width of the update area, which is the clustering center position of the winning neuron. The improved Δ*c_ij_* is:(3)$$\Delta c_{j}=\{\begin{array}{c} \eta h_{j}\left(x-c_{j}\right)\left(2-p_{j}\right);y_{j}=1\\ \eta h_{j}\left(x-c_{j}\right)p_{j}\frac{\left(2-p_{j}\right)}{\left(1-p_{j}\right)};y_{j}=0 \end{array}$$

After using random samples for iterative calculations until the clustering centers of neurons meet the accuracy requirement, the clustering centers of the historical prediction samples are finally obtained as the radial center of the RBF surface roughness prediction model.

### 2.3. Establishment of Prediction Model of Superalloy Surface Roughness Based on Radial Basis Function Neural Network

The purpose of the RBF neural network training is to find a nonlinear function that can satisfy the relationship between abrasive belt grinding parameters as the input and the surface roughness after superalloy grinding as output parameters. Therefore, to establish a model for predicting the surface roughness of abrasive belt grinding based on the RLSOM-RBF neural network, we must first determine each layer’s input neuron node and output neuron node in the RBF neural network and parameter selection.

#### 2.3.1. Input and Output Neurons

The existing processing experiences of the surface, belt type, cooling conditions, machine parameters, grinding pressure, feed rate, belt speed, feed step, etc., will affect the surface roughness. However, if all the influencing factors are used as input neurons, the network complexity of the prediction model will increase, resulting in a slower learning speed. The traditional surface roughness prediction formula is $Ra=KF^{\beta }v_{f}^{\gamma }v_{s}^{\delta }$:(4)$$r_{j}=\{\begin{array}{c} h_{j};q_{j}=1\\ -h_{j};q_{j}=0 \end{array},h_{j}=\exp\left(\frac{\left\|c_{j}-c^{*}\right\|}{2\sigma^{2}}\right)$$

The grinding pressure, linear belt velocity, and feed speed present a specific functional relationship between the three parameters. This paper uses these grinding parameters as input layer neurons and the surface roughness under these grinding parameters as output neurons. Therefore, the neural network model established for the surface roughness prediction of high-temperature alloy belt grinding is shown in Figure 2.

#### 2.3.2. Selection of Hidden Layer Parameters

The function approximation can theoretically be achieved for any input and output samples when the radial basis neural network is used. However, the accuracy of the clustering centers of RBF neurons can be influenced. Therefore, the gradient descent method is used to train the overall parameters of the RBF neural network surface roughness model. On this basis, the relationship between the input grinding pressure, linear belt velocity, feed rate information, and predicted surface roughness is fitted. The overall parameter training includes modifying the radial basis center of the prediction model, training the broad band *σ* of the Gaussian function, and the connection weight *ω* from the hidden layer to the output layer. The loss function for constructing the RBF surface roughness prediction model is:(5)$$E=\frac{1}{2}\overset{N}{\underset{k=1}{\sum }}e_{k}^{2},e_{k}=y^{k}-y_{m}^{k}=y^{k}-\overset{N}{\underset{j=1}{\sum }}\omega_{j}R_{j}^{k}$$
(6)$$R_{j}^{k}=\exp\left(\frac{-\|x_{k}-c_{j}\|}{2\sigma_{j}^{2}}\right)$$

In the formula, *E* and *e_k_* are the total error and sample error, respectively; *R_j_* is the output of the *j* neuron in the competition layer; *x_k_* is the *k* sample; *N* is the number of pieces, and *y_k_^m^* are the target surfaces of the *k* sample’s roughness and predicted surface roughness calculated by the model, respectively. The iteration formulas for solving *c*, *σ*, and *ω* according to the gradient descent method are:(7)$$\{\begin{array}{c} \Delta c_{ji}=\frac{\partial E}{\partial c_{ji}}=\overset{N}{\underset{k=1}{\sum }}\left(y^{k}-y_{m}^{k}\right)\omega_{j}R_{j}^{k}\frac{x_{i}^{k}-c_{ji}}{\sigma_{j}^{2}}\\ c_{ji}\left(t\right)=c_{ji}\left(t-1\right)+\eta_{c}\Delta c_{ji}+\alpha_{c}\left(c_{ji}\left(t-1\right)-c_{ji}\left(t-2\right)\right) \end{array}$$
(8)$$\{\begin{array}{c} \Delta \sigma_{j}=\frac{\partial E}{\partial \sigma_{j}}=\overset{N}{\underset{k=1}{\sum }}\left(y^{k}-y_{m}^{k}\right)\omega_{j}R_{j}^{k}\frac{\|x_{k}-c_{j}{\|}^{2}}{\sigma_{j}^{2}}\\ \sigma_{j}\left(t\right)=\sigma_{j}\left(t-1\right)+\eta_{\sigma }\Delta \sigma_{j}+\alpha_{\sigma }\left(\sigma_{j}\left(t-1\right)-\sigma_{j}\left(t-2\right)\right) \end{array}$$
(9)$$\{\begin{array}{c} \Delta \omega_{j}=\frac{\partial E}{\partial \omega_{j}}=\overset{N}{\underset{k=1}{\sum }}\left(y^{k}-y_{m}^{k}\right)R_{j}^{k}\\ \omega_{j}\left(t\right)=\omega_{j}\left(t-1\right)+\eta_{\omega }\Delta \omega_{j}+\alpha_{\omega }\left(\omega_{j}\left(t-1\right)-\omega_{j}\left(t-2\right)\right) \end{array}$$

The formula represents the *t* iteration; *η* and *α* are the learning rates of different iterations.

## 3. Abrasive Belt Grinding Experiment and Experimental Results

The selection of neural network training samples will directly affect the prediction accuracy. Due to the limited surface roughness collected in the experiment, the traditional algorithm evaluation method has certain values of randomness and chance. Therefore, this paper adopts the surface roughness prediction model for superalloy abrasive belt grinding with the ten-fold cross validation method; that is, every ten groups of data are used as a dataset, whereby nine groups of one dataset are taken as training data for learning and one group is taken as experimental test data for the experiment. The average relative error of the 10 sets of prediction results is used as the model accuracy to evaluate the accuracy of the algorithm. For the sample data, the BP neural network, the SOM-RBF neural network, and the RLSOM-RBF neural network were used to carry out simulation experiments on MATLAB using the ten-fold crossover method. The experiment and simulation ideas are shown in Figure 3. The experiment was carried out on the precision numerical control belt grinding experiment platform, and the main relevant parameters are shown in Table 1. The Form Talysurf Series II surface roughness profiler was used to measure the surface roughness. The measurement range is 2 mm, the resolution is 32 nm/2 mm, and the parameter measurement accuracy is 2% ± 6 nm. In the experiment, the rectangular GH4169 nickel-based superalloy was used, and its size was 92 mm × 62 mm × 12 mm. The chemical composition and mechanical properties are shown in Table 2. The abrasive belt was ceramic based (XK870K, P120).

Before the single-factor experiment, an exploratory experiment was conducted to determine the appropriate parameter selection range, and to detect the initial surface roughness of the test piece. The selected processing parameters, which contain the abrasive belt speed, feed rate, and grinding pressure, are the ones that directly affect the abrasive belt grinding processing, so these three parameters are selected as input parameters. The parameter selection is shown in Table 3, which mainly focuses on the surface quality of the processing under the combination of smaller and larger processing parameters. In the case of the smaller combination of processing parameters, the problem of empty grinding occurred, as shown in Figure 4a. Apparent surface defects appeared in the case of the larger processing parameter combination, as shown in Figure 4b.

Under the “0-0-0” processing parameter combination, the average initial surface roughness (*Ra*) of the test piece was measured as 5.570 μm. In order to reduce the influence of the initial surface quality on the surface roughness value in the subsequent experiment, the test piece was processed by pre-grinding in the experiment. After measuring, the surface roughness (*Ra*) of the test piece after pre-grinding was 2.230 ± 0.150 μm.

After the exploratory experiment, an abrasive belt grinding experiment was conducted on GH4169 superalloy material by setting different linear belt speeds, feed speeds, and grinding pressures. In order to minimize the influence of processing errors and measurement errors on the surface roughness measurement values, five grinding marks were processed under the same combination of processing parameters. Each grinding mark was divided into three areas. Each area is measured three times. That is, there were 40 measuring point data for each group of processing parameter combinations. The statistically related abrasive belt grinding data and the surface roughness samples after grinding are shown in Table 4; see Appendix A for complete data.

## 4. Discussion and Analysis

### 4.1. Simulation Results

MATLAB simulated the related grinding parameters and the surface roughness after grinding to obtain the predicted surface roughness under different prediction models. Then, the relative errors between them were compared. The relative error is shown in Table 5, and the relative error graph is shown in Figure 5.

### 4.2. Simulation Analysis

From the prediction curve, we can see that the neural network has a good effect on the prediction of the surface roughness of the high-temperature alloy belt grinding. Compared with the traditional BP neural network prediction model, the RBF neural network prediction model has higher accuracy. Here, based on the simulation analysis of the traditional SOM-RBF neural network prediction model and the improved RLSOM-RBF neural network prediction model, the results show that the accuracy of the RLSOM-RBF neural network prediction model is higher than the traditional SOM-RBF prediction accuracy. Through the analysis of relative errors, we can see that the maximum relative error of the BP prediction model is 8.1% and the minimum relative error is 1.3%; the maximum relative error of the SOM-RBF prediction model is 7.5% and the minimum relative error is 1%. -RBF the maximum relative error of the prediction model is 3.1%, and the minimum relative error is 0%.

In addition, using the experimental data measured in the previous high-temperature alloy belt grinding experiment with the large number of samples, MATLAB was used to perform error surface analysis and model errors (training error, training situation error, training) with three different prediction models. From the error surface, the BP neural network prediction model has the highest error, the RL-SOMRBF neural network prediction model has the lowest error, and the partial values from the SOM-RBF and RL-SOMRBF neural network prediction models are more obvious, as shown in Figure 6.

As can be seen from Figure 7, from the model error point of view, the RL-SOMRBF prediction model has a minor error of about 0.00178, followed by the SOMRBF prediction model with about 0.145; finally, the BP prediction model has an error of about 0.3741. It can be seen that for the prediction of the surface roughness of abrasive belt grinding, the RBF neural network prediction model is superior to the BP neural network prediction model. In addition, by improving the RBF training center method, the prediction accuracy of the RBF neural network can be improved.

After the MATLAB training was completed, neural network training results graphs for three different surface roughness prediction models were obtained. From Figure 8a, we can see that the mean squared error (mse) of the BP model is significant, indicating that the accuracy of the model is not very high. As the model reaches the best accuracy, there are significant deviations in the training curve, test curve, and verification curve, indicating that training overfitting occurred during the process.

Compared with the BP neural network surface roughness prediction model, the SOM-RBF and RLSOM-RBF neural network prediction models have no overfitting. The training results are better, but when the RLSOM-RBF is compared to the traditional SOM-RBF training method, the training time is longer, which solves the problems of easily falling into the optimal local solution and weak generalization ability, as shown in Figure 8b,c.

If the neural network samples are input into the network, the samples will be divided into three categories by default: Training samples, verification samples, and test samples. The default value of the validation check is 6, which means that as the network uses the training samples for training, it is confirmed that the error curve of the models does not decline for six consecutive iterations. With the training of the network, it is confirmed that the error of the sample has basically not decreased or even increased, so there is no need to train the network again. If the training continues, the test sample is used to test the network, and there will be no improvement in the error of the test sample, and even overfitting will occur.

In Figure 9a, one can see that the gradient training of the BP model continues to decline, but the variable mu decreases with the gradient. The accumulated error reaches the maximum before the training is completed. The minimum error does indeed start to appear, and when the number of iterations is 9, the training parameters do not change; that is, they fall into the optimal local solution.

For the SOM-RBF model, the gradient training of the model drops more slowly than the BP model, and as the training progresses, the variable mu first decreases to a minimum, then the cumulative error increases and then decreases, but does not decrease to the smallest error. This shows that the generalization ability of the model is weak, and the optimal solution of the model is reached when the number of iterations is 9, as shown in Figure 9b.

For the RLSOM-RBF neural network model, the gradient descent trend of this model is slower than that of the other two models, and the mu value is also consistent with the gradient descent trend. The optimal solution is reached when the number of iterations is 46. Compared with the other two models, the optimal local solution and weak generalization ability do not occur, as shown in Figure 9c.

In the final prediction of network training, a simple regression analysis is used. Part of the data are used for training, part of the data are used to confirm the training, and the rest of the information is used for testing and the final overall situation, representing training samples, verification samples, test samples, and overall prediction results, respectively. The closer the R-value is to 1, the better the model and the higher the prediction accuracy.

From Figure 10a, during the fitting process of the BP neural network prediction model, we can see that the fitting effect of the model is better during the training process, and the distribution is more uniform. In the confirmation of the training situation, one can see the fitting. The effect is very close to the target, as shown in a and b in Figure 10a. Still, in the process of verifying the sample, the fitting effect of the target roughness and output roughness in general, as well as some discrete points, are far from the fitting line, as shown in Figure 10a. Therefore, the overall fitting effect is better in the interval of the roughness of 0.2 μm–0.4 μm. In the interval of 0.4 μm–0.6 μm, the output roughness and the predicted roughness are quite different. As a result ofthe excessive speed, the overall R-value is 0.765.

During the fitting process of the SOM-RBF neural network prediction model, we can see that during the training process, as shown in Figure 10b, the fitting effect of the model is general, and the predicted roughness value is close to the output roughness score. It can be seen in the confirmation training situation in Figure 10b that the output roughness of the roughness prediction model is relative to the predicted roughness. In the subsequent process of verifying the sample, the fitting effect of the target roughness and the output roughness is generally good, and some discrete points are far away from the fitted line, as shown in Figure 10c. Therefore, the overall fitting effect is better in the roughness interval of 0.3 μm–0.5 μm. At this time, the model may reach the optimal local solution. When the roughness exceeds 0.5 μm, the output roughness and predicted roughness is quite different, and the overall R-value is 0.9238.

By analyzing the errors and training results of the three algorithm models it can be determined that the RLSOM-RBF neural network prediction model is more effective than the other two (BP model and SOM-RBF model) in predicting the surface roughness of superalloy under abrasive belt grinding. This shows that reinforcement learning is not only effective for the modification of the traditional RBF neural network training method, but also shows adaptability in predicting the surface roughness of superalloy under abrasive belt grinding. The characteristics of superalloys make them prone to work hardening, inversion of the relationship between the abrasive grains and the workpiece, abrasive wear, etc., in the process of grinding superalloys by abrasive belt, resulting in the BP model and the SOM-RBF model not being able to fit the processing conditions well. In addition, the high nonlinearity and high coupling characteristics of abrasive belt grinding increase the difficulty of prediction in these two models.

## 5. Conclusions

First of all, the use of reinforcement learning (RL) improved the SOM method. Three layers of competition according to the competitive reward mechanism structure of the neural network prediction data clustering model were used. Then, by using the improved stochastic gradient reinforcement training model parameters and updating the clustering center, eventually the competitive clustering center of the biggest set of neurons became the center of the radial basis, which improved the traditional radial basis neural network training method and allowed it to easily to fall into the optimal local solution. Second, through the design of crossing one hundred percent of the abrasive belt grinding tests, every ten groups of data were considered a dataset wherein nine groups were considered the training data and one set was the experimental data. We recorded the relative error value and the average relative error of the prediction 10 times in order to obtain the model accuracy. Through comparisons of the three kinds of algorithms, one can see that the BP neural network prediction model of surface roughness displayed an average relative error of 3.55%, the SOM-RBF neural network prediction model of surface roughness gave a poor relative average of 3.35%, and the average relative error of the surface roughness prediction model of the RLSOM-RBF neural network was only 1.73%, indicating that it is effective in improving the traditional training method through reinforcement learning. Finally, by using much of the abrasive belt grinding experiment data as the training sample and importing the three surface roughness prediction models of the neural network, we were able to analyze the error curved surface model, the model error, the training results, and the changes in the parameters. Through this analysis, we determined that the RLSOM-RBF model prediction is better than the SOM-RBF model and is superior to the BP model. By analyzing the R values fitted by the three prediction models, we can see that the R-value of the RLSOM-RBF model is 0.996, the R-value of the SOM-RBF neural network model is 0.923, and the R-value of the BP neural network model is 0.765. It shows that using the reinforcement algorithm to improve the traditional radial basis training center is very effective because it can solve highly nonlinear and highly coupled characteristics and can be used to predict the surface roughness of superalloy belt grinding.

However, the accuracy of the model proposed in the article can be further improved. In addition, the generalization ability of the model is weak, the portability is poor, and the robustness needs to be further improved.

## Figures and Tables

**Figure 1 materials-14-05701-f001:**
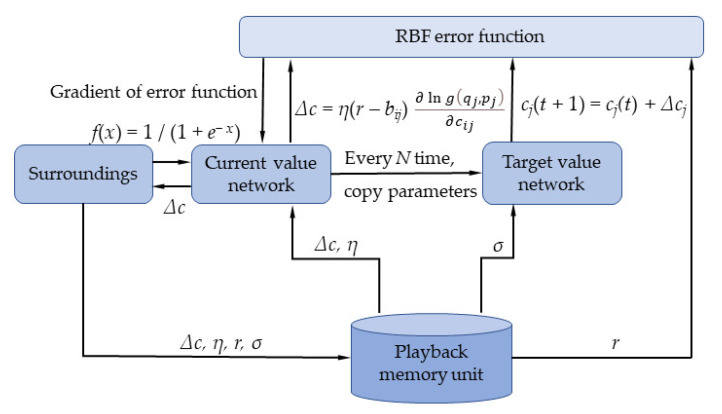
Algorithm training flowchart.

**Figure 2 materials-14-05701-f002:**
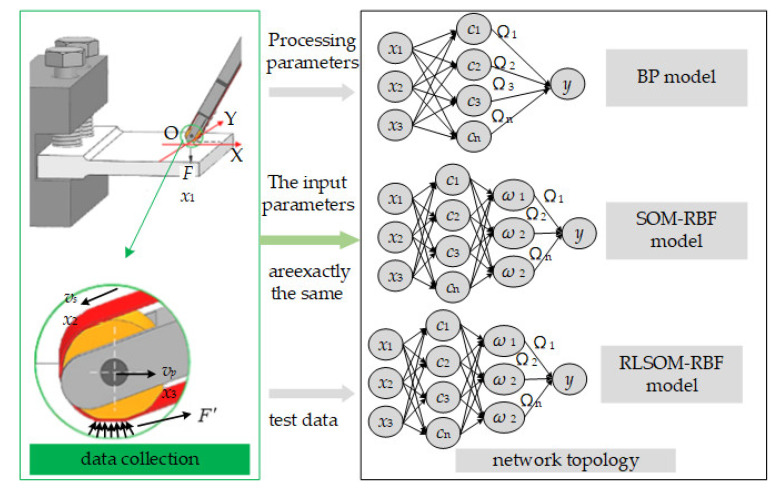
Neural network model for surface roughness prediction.

**Figure 3 materials-14-05701-f003:**
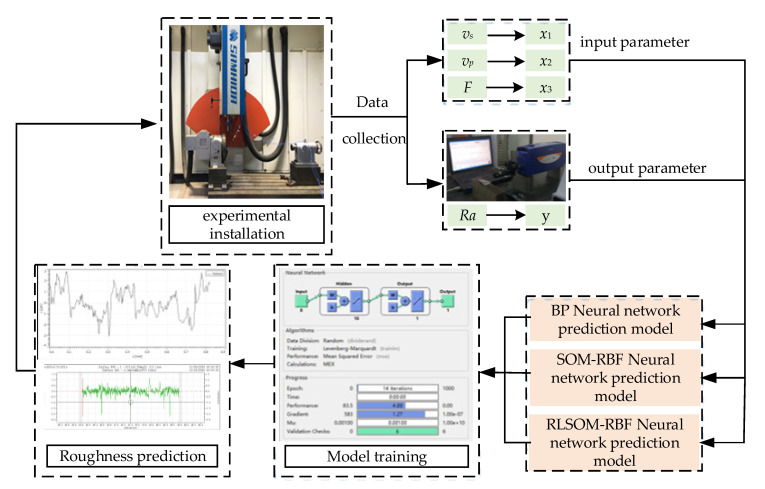
Flowchart of neural network for surface roughness prediction.

**Figure 4 materials-14-05701-f004:**
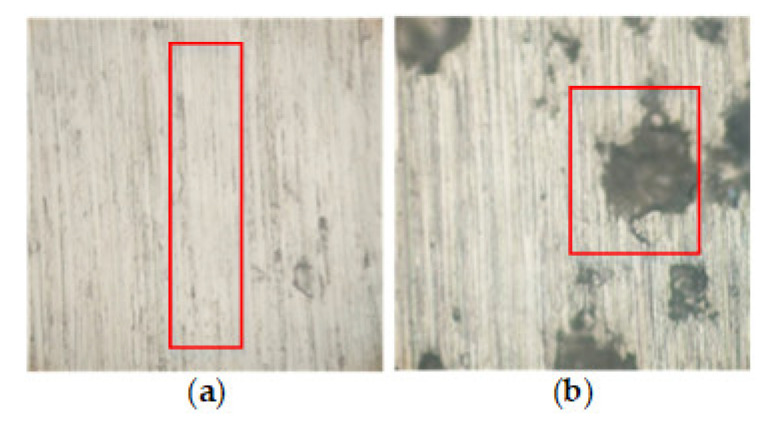
(**a**) Empty grinding; (**b**) apparent surface defects.

**Figure 5 materials-14-05701-f005:**
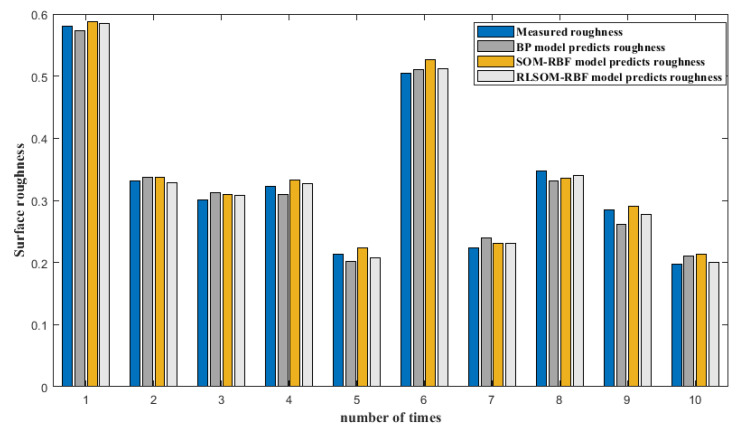
Comparison between experiment and prediction.

**Figure 6 materials-14-05701-f006:**
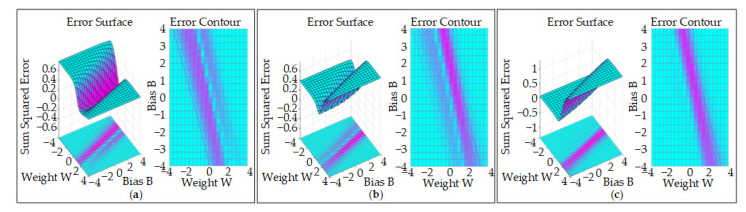
The error surface of neural network prediction models: (**a**) BP model; (**b**) SOM-RBF model; (**c**) RLSOM-RBF model.

**Figure 7 materials-14-05701-f007:**
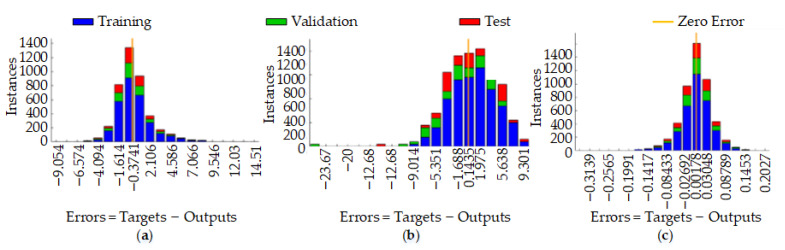
Prediction model accuracy error chart: (**a**) BP model; (**b**) SOM-RBF model; (**c**) RLSOM-RBF model.

**Figure 8 materials-14-05701-f008:**
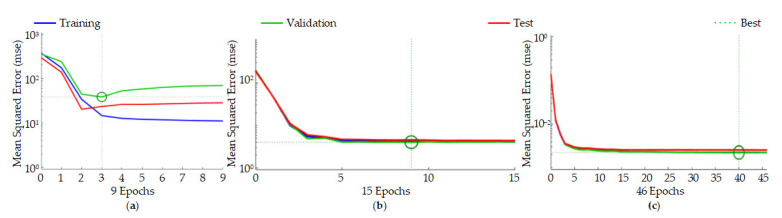
Training results: (**a**) BP model; (**b**) SOM-RBF model; (**c**) RLSOM-RBF model.

**Figure 9 materials-14-05701-f009:**
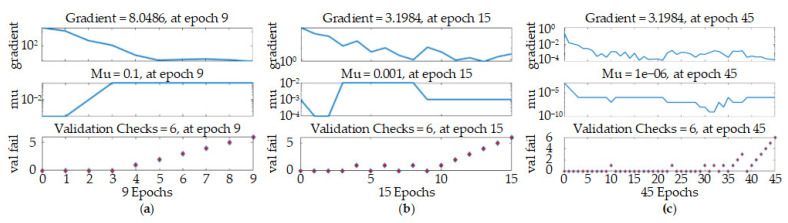
Model training parameter changes: (**a**) BP model; (**b**) SOM-RBF model; (**c**) RLSOM-RBF model.

**Figure 10 materials-14-05701-f010:**
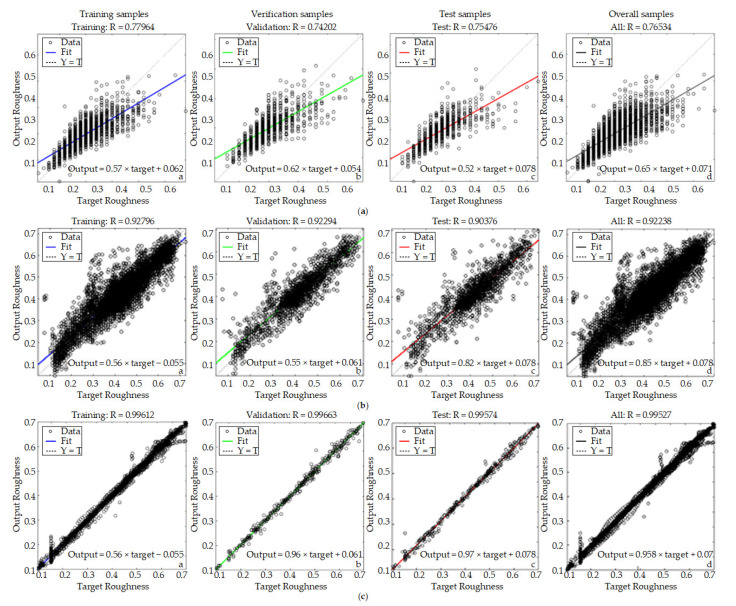
Model fitting diagram: (**a**) BP model; (**b**) SOM-RBF model; (**c**) RLSOM-RBF model.

**Table 1 materials-14-05701-t001:** The main relevant parameters of the experiment platform.

X/Y/Z Axis Positioning Accuracy	X/Y/Z Axis Repeat Positioning Accuracy	A-Axis Positioning Accuracy	Surface Roughness Ra
0.015 mm	0.01 mm	0.01°	0.1 μm~0.8 μm

**Table 2 materials-14-05701-t002:** Chemical composition and mechanical properties of GH4169 superalloy.

**Chemical Composition (%)**						
Ni	Cr	Al	Mo	Ti	C	Nb
52.30	18.90	0.52	3.08	1.06	0.04	5.30
Mn	Si	Cu	Ta	Co	P	Fe
<0.20	<0.20	<0.20	<0.20	<0.20	<0.20	remaining
**Mechanical Properties**						
Elastic modulus *E*/GPa	Thermal conductivity *λ*/W.m	Elongation *δ*/%	Hardness HB (Room temperature)	Impact value *aK*/(J.cm^−2^)	Shrinkage rate *ψ*/%	Melting point /°C
205	14.65	15	346–450	573	41	1260–1320

**Table 3 materials-14-05701-t003:** Processing parameter selection table in the exploratory experiment.

Abrasive Belt Speed (m/s)	Feed Rate (m/s)	Grinding Pressure (N)
0	0	0
5	0.01	5
10	0.02	10
18	0.04	20
26	0.06	30
32	0.08	40
40	0.10	50

**Table 4 materials-14-05701-t004:** Abrasive belt grinding experiment record table.

Serial Number	Abrasive Belt Speed (m/s)	Feed Rate (m/s)	Grinding Pressure (N)	Measured Surface Roughness (μm)
1	10	0.02	10	0.581
2	10	0.04	20	0.332
3	10	0.06	30	0.301
4	18	0.02	20	0.322
5	18	0.04	30	0.214
6	18	0.06	10	0.505
7	26	0.02	30	0.224
8	26	0.04	10	0.347
9	26	0.06	20	0.284
10	32	0.08	40	0.198

**Table 5 materials-14-05701-t005:** Table of relative error of different prediction models.

Expected Output	BP		SOM-RBF		RLSOM-RBF	
	Predictive Value (μm)	Relative Error	Predictive Value (μm)	Relative Error	Predictive Value (μm)	Relative Error
0.581	0.473	−1.3%	0.587	1%	0.584	0.52%
0.332	0.337	1.5%	0.337	1.5%	0.329	−0.90%
0.301	0.312	3.7%	0.310	2.9%	0.308	2.3%
0.322	0.310	3.7%	0.333	3.4%	0.327	−1.5%
0.214	0.202	−5.6%	0.224	4.6%	0.208	−2.8%
0.505	0.511	1.2%	0.526	5.1%	0.512	1.4%
0.224	0.239	6.7%	0.231	3.1%	0.231	3.1%
0.347	0.332	−4.3%	0.355	2.3%	0.340	2%
0.284	0.261	−8.1%	0.290	2.1%	0.278	−2.8%
0.198	0.210	6.1%	0.213	7.5%	0.198	0
Mean relative error		3.55%		3.35%		1.732%

## Data Availability

Not applicable.

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
