# Peer review of "Prediction of Surface Roughness of Abrasive Belt Grinding of Superalloy Material Based on RLSOM-RBF"

_materials, 2021, doi:10.3390/ma14195701_

Round 1
Reviewer 1 Report
Suggestions for improving the article are as follows:
- Technical suggestion: Most figures, almost all, are not legible and clear. You need to prepare figures in better quality (better resolution).
- The article is classified as a "Review". This is not a review manuscript.
- In the last paragraph of the Introduction section, a general overview of previous research is given. A critical analysis of previous research needs to be more detailed. Add more details in the corrected article.
- Add another paragraph at the end of the Introduction section (after the existing last paragraph). Based on the previous critical analysis, explicitly define the goal of your research, as well as the scientific hypothesis. Also, emphasize scientific contribution and scientific benefits of your research.
- The scientific contribution and scientific benefits are not clearly highlighted in the article. This should be done in the Introduction section, but also in all other places when the authors think it can be pointed out.
- Very little data on the experiment is given. They must be significantly supplemented. Authors must provide detailed information on materials (mechanical, thermal and physical characteristics). Missing data on: machine tools, tools, accessories, machining parameters, etc. It is also necessary to explain the choice. Also, the processing should be further briefly described.
- The choice of processing parameter levels (speed, feed rate, pressure) should be explained separately. Why did you choose these levels of processing parameters? Why these levels were representative of your research? Why didn't you consider other input parameters? Everything previously should be elaborated in detail in the article.
- You must provide detailed information about the workpiece, its geometry, etc. The initial surfaces roughness to be treated must also be shown. That is very important to show.
- Do you measure Ra? I did not notice that this is emphasized anywhere in the article except in equation 1.
- Measurement data are not complete. This data must be detailed as it is input data. On which device did you perform the measurement and in what way. Also show detailed information about the measurement process. The parameters of the measurement process should also be given (cut-off length, stylus radius, length of evaluation, etc.)
- A major drawback is that other errors (processing errors, measurement errors, etc.) and their impact on the results have not been analyzed and discussed. These errors have a direct and large impact on the results. Can you estimate the measurement uncertainty of the obtained roughness measurement results.
- The physics or the science behind the experiments needs to be clarified with the interpretation of the results. This must be done because of the great impact on the results. The results must be scientifically discussed and compared.
- In the conclusions, state the limitations of your research and the directions of future research.
- General comment: I ask the authors to pay attention to the following:
"Aims
...... Our aim is to encourage scientists to publish their experimental and theoretical results in as much detail as possible. Therefore, there is no restriction on the length of the papers. The full experimental details must be provided so that the results can be reproduced.
Materials provides a forum for publishing papers which advance the in-depth understanding of the relationship between the structure, the properties or the functions of all kinds of materials. Chemical syntheses, chemical structures and mechanical, chemical, electronic, magnetic and optical properties and various applications will be considered."
Previously missing in this study.
Reviewer 2 Report
The presented article entitled Prediction of surface roughness of abrasive belt grinding of superalloy material based on RLSOM-RBF is primarily devoted to the prediction of grinding surface roughness using neural networks. The article is based on the analysis of neural networks and not on the research of technology and its impact on the surface or physical properties of the surface. I have a few comments on the article:
- What makes your article new? What new information do you bring?
- 2. Figure 6, 7, 8 and especially Figure 9 is illegible, it is necessary to increase the quality of resolution
- The article lacks verification of the surface roughness model even under other grinding conditions, because at this stage it is possible to consider its validity only under the conditions given in Table 1.
- Overall, the article is based on the application of neural networks without a deeper analysis of the results obtained from a technological and material point of view.
Reviewer 3 Report
The review report it´s send in attach pdf file

Round 2
Reviewer 1 Report
The article has been supplemented and updated. I suggest accepting the article.
Reviewer 2 Report
The submitted scientific article, after editing by the authors, has increased its quality and can be published.